# Unlike its Paralog LEDGF/p75, HRP-2 Is Dispensable for MLL-R Leukemogenesis but Important for Leukemic Cell Survival

**DOI:** 10.3390/cells10010192

**Published:** 2021-01-19

**Authors:** Siska Van Belle, Sara El Ashkar, Kateřina Čermáková, Filip Matthijssens, Steven Goossens, Alessandro Canella, Courtney H. Hodges, Frauke Christ, Jan De Rijck, Pieter Van Vlierberghe, Václav Veverka, Zeger Debyser

**Affiliations:** 1Laboratory for Molecular Virology and Gene Therapy, Department of Pharmaceutical and Pharmacological Sciences, KU Leuven, 3000 Leuven, Belgium; siska.vanbelle@kuleuven.be (S.V.B.); sara.elashkar@gmail.com (S.E.A.); Alessandro.Canella@osumc.edu (A.C.); frauke.christ@kuleuven.be (F.C.); janderijck1@gmail.com (J.D.R.); 2Institute of Organic Chemistry and Biochemistry, Academy of Sciences of the Czech Republic, Flemingovo nam. 2, 166 10 Prague, Czech Republic; Katerina.Cermakova@bcm.edu (K.Č.); vaclav.veverka@uochb.cas.cz (V.V.); 3Department of Biomolecular Medicine, Ghent University, 9000 Ghent, Belgium; Filip.Matthijssens@UGent.be (F.M.); Pieter.VanVlierberghe@UGent.be (P.V.V.); 4Cancer Research Institute Ghent (CRIG), 9000 Ghent, Belgium; Steven.Goossens@UGent.be; 5Department of Diagnostic Sciences, Ghent University, 9000 Ghent, Belgium; 6Department of Molecular & Cellular Biology, Center for Precision Environmental Health, and Dan L Duncan Comprehensive Cancer Center, Baylor College of Medicine, Houston, TX 77030, USA; Courtney.Hodges@bcm.edu; 7Department of Cell Biology, Faculty of Science, Charles University, 128 00 Prague, Czech Republic

**Keywords:** leukemia, molecular cell biology, protein complex, protein-protein interaction, nuclear magnetic resonance (NMR), animal model, cell culture, hematopoietic stem cell, cell proliferation

## Abstract

HDGF-related protein 2 (HRP-2) is a member of the Hepatoma-Derived Growth Factor-related protein family that harbors the structured PWWP and Integrase Binding Domain, known to associate with methylated histone tails or cellular and viral proteins, respectively. Interestingly, HRP-2 is a paralog of Lens Epithelium Derived Growth Factor p75 (LEDGF/p75), which is essential for *MLL*-rearranged (*MLL*-r) leukemia but dispensable for hematopoiesis. Sequel to these findings, we investigated the role of HRP-2 in hematopoiesis and *MLL*-r leukemia. Protein interactions were investigated by co-immunoprecipitation and validated using recombinant proteins in NMR. A systemic knockout mouse model was used to study normal hematopoiesis and MLL-ENL transformation upon the different HRP-2 genotypes. The role of HRP-2 in *MLL*-r and other leukemic, human cell lines was evaluated by lentiviral-mediated miRNA targeting HRP-2. We demonstrate that MLL and HRP-2 interact through a conserved interface, although this interaction proved less dependent on menin than the MLL-LEDGF/p75 interaction. The systemic HRP-2 knockout mice only revealed an increase in neutrophils in the peripheral blood, whereas the depletion of HRP-2 in leukemic cell lines and transformed primary murine cells resulted in reduced colony formation independently of *MLL*-rearrangements. In contrast, primary murine HRP-2 knockout cells were efficiently transformed by the MLL-ENL fusion, indicating that HRP-2, unlike LEDGF/p75, is dispensable for the transformation of MLL-ENL leukemogenesis but important for leukemic cell survival.

## 1. Introduction

Mixed Lineage Leukemia (MLL)-rearranged leukemia is an aggressive, genetically distinct subset of acute leukemia. These rearrangements, involving the *11q23* locus, are more frequently observed in infants, for which percentages range between 35% and 70% according to different studies [1,2,3,4,5]. Despite multiple efforts to develop a precision treatment, *MLL-*rearranged (*MLL*-r) acute leukemia remains associated with poor prognosis in children and adults, mostly due to tumor relapse, resulting in low 5-year survival rates [6]. The hallmark of this cancer is the translocation of the gene encoding MLL (*MLL1* or *KMT2A*) histone methyltransferase with one of more than 80 different partner genes, leading to the formation of oncogenic fusion proteins (MLL-FPs) [7]. The majority of MLL fusion partners are transcription regulators and promote the aberrant recruitment of the transcription machinery to the MLL target genes such as *HOX* genes (for review see Slany RK. [8]). To regulate gene expression, unstructured N-terminal motifs found in MLL form a ternary complex with menin and the integrase binding domain (IBD) of the p75 splice variant of Lens Epithelium Derived Growth Factor (LEDGF/p75). While LEDGF/p75 tethers the MLL-FPs to target genes, the sole molecular requirement for menin as an oncogenic cofactor in *MLL-*r leukemogenesis is to stabilize the interface of LEDGF/p75 with MLL-FPs [9]. Formation of the ternary complex is crucial for *MLL*-r leukemogenesis [9,10,11]. Importantly, LEDGF/p75 is dispensable for normal hematopoiesis, but required for the development of mixed lineage leukemia in mice [12].

HDGF, HDGF-related proteins 1 to 4 (HRP1-4) and LEDG, belong to the Hepatoma Derived Growth Factor (HDGF) family of proteins. The members are differentially expressed in both normal and pathological tissues, and their functions are not completely understood [13,14]. They are characterized by a high similarity in their N-terminal Homologue to Amino Terminus of HDGF (HATH) region [15]. This domain encompasses a PWWP-domain that specifically interacts with methylated lysines in histone tails [16]. Of note, HRP-2 and LEDGF/p75 are the only two members of this family harboring an IBD at their C-terminal end. LEDGF/p75 is a transcriptional co-activator that tethers IBD-binding proteins to the chromatin. Previous research indicated a role for LEDGF/p75 in DNA repair [17], cell survival and stress response [18,19,20], and various cancers [21,22,23,24,25].

The IBD domain of LEDGF/p75 is involved in the binding to the MLL-menin complex [9] and also known to interact with other cellular proteins including JPO2 [26], PogZ [27], ASK [28], IWS1 [29] and MED1 [30]. The affinity to the IBD domain is regulated by phosphorylation of the IBD binding motif (IBM) present on these binding partners [30]. In co-immunoprecipitation experiments PogZ, JPO2 and IWS1 were confirmed as HRP-2 binding partners [27,31]. 

In 2003, we reported that LEDGF/p75, next to cellular proteins, interacts with HIV integrase through the IBD [32,33]. By tethering the integration complex to the chromatin, LEDGF/p75 supports lentiviral integration and replication [32,34,35,36]. HRP-2 overexpression substitutes for the LEDGF/p75 function upon the depletion of the latter, presumably the consequence of the lower affinity of HIV-1 integrase for HRP-2 than for LEDGF/p75 [33,37]. Besides its role in HIV infection, HRP-2 has been linked to DNA repair by homologous recombination [38]. In addition, both HRP-2 and LEDGF/p75 were recently reported as key factors allowing RNA polymerase II to overcome the nucleosome-induced barrier to transcription elongation by taking over the FACT (facilitates chromatin transcription) complex role in differentiation to myotubes [39]. A myogenic function was also suggested by Zhang X. et al. [40]. HRP-2 is reported to be overexpressed in up to 40% of hepatocellular carcinoma cells where it promotes cancer cell growth in vitro as well as in vivo [31], whereas a reduction in HRP-2 was reported as a poor prognostic factor in Helicobacter pylori induced gastric cancer [41].

In light of this putative oncogenic activity, we compared HRP-2 with LEDGF/p75 in the context of hematopoiesis and *MLL-*r leukemia. Our data show that HRP-2 directly interacts with MLL in a menin-independent way. We report a systemic HRP-2 knockout mouse model with some postnatal mortality and modest changes in blood counts in the knockout survivors. In contrast to LEDGF/p75, HRP-2 appears to be dispensable for *MLL-*r leukemogenesis but required for leukemic cell survival. 

## 2. Materials and Methods

### 2.1. Viral Vector Transductions and Generation of Stable Cell Lines 

Viral vector productions were performed as previously described [12,42]. Titer units (TU) were determined by a p24 ELISA test (Fujirebio, Belgium). Human cell lines were transduced in a 1:1 volume ratio with concentrated lentiviral supernatants. Murine cells were transduced with pMSCV-based vectors as previously described [12]. Forty-eight hours post-transduction, cells were selected with puromycin at 1 µg/mL. 

### 2.2. Protein Purification

All proteins were expressed in *E. coli* Rosetta2 (DE3), grown on Lysogeny Broth medium and supplemented with 10 mg/mL ampicillin. Flag-tagged LEDGF/p75 purification was performed as described before [43]. MBP-tagged HRP-2_470–552_ and LEDGF/p75_325–530_ were purified similarly as described for the latter in [44]. MLL_1–160_-GST was purified as described in [45]. Expression and purification of LEDGF/p75_345–426_ was described earlier [46]. Identical conditions were used for expression and purification of HRP-2_469–549_. Protein fractions were analyzed by SDS-PAGE and Coomassie stain. Peak fractions were pooled. 

### 2.3. AlphaScreen Assay

The AlphaScreen assay was performed as described before [30]. Flag-LEDGF/p75 (0.3 nM) was preincubated with MLL_1–160_–GST (10 nM) before titration of MBP-LEDGF/p75_325–530_, MBP-HRP-2_470–552_ or MBP alone at the indicated concentrations. 

### 2.4. Co-Immunoprecipitation (IP)

Six million HEK293T cells were plated in a 8.5 cm petri dish and transfected with 20 µg of each indicated plasmid (three petri dishes/condition) and lysed as described in [30]. For IP experiments with MI-2 and MI-538 (MedChem Express), respectively 100 μM and 50 µM of the compound (or DMSO control) was added during lysis and overnight incubation. Immunoprecipitated protein was eluted with SDS-PAGE loading buffer and visualized by western blotting.

### 2.5. Peptide Synthesis

The peptides used in this study were synthesized by solid-phase synthesis in the Laboratory of Medicinal chemistry, IOCB, ASCR (Prague, Czech Republic).

### 2.6. NMR Spectroscopy

NMR spectra were acquired at 25 °C on the 850 MHz Bruker Avance spectrometer equipped with a triple-resonance (^15^N/^13^C/^1^H) cryoprobe. For structure determination, the sample volume was 0.35 mL, with a concentration of 500 μM HRP-2_469–549_ in the NMR buffer (25 mM TRIS pH 7.0, 150 mM NaCl, 1 mM TCEP), 5% D_2_O/95% H_2_O. The sequence-specific backbone and side-chain resonance assignment were obtained using a series of standard triple-resonance spectra (HNCO, HN(CA)CO, HNCACB, CBCA(CO)NH, HBHA(CO)NH, CCC(CO)HN and HCCH-TOCSY [47,48]). ^1^H-^1^H distance constraints for structural determination were obtained from intensities of NOE cross peaks in the 3D ^15^N/^1^H NOESY-HSQC and ^13^C/^1^H NOESY-HMQC spectra that were acquired using a NOE mixing time of 100 ms.

The families of converged structures were initially calculated in Cyana 3.98 using the combined automated NOE assignment and structure determination protocol [49]. In addition, backbone torsion angle constraints, generated from assigned chemical shifts using the program TALOS+ [50] were included in the calculations. Subsequently, five cycles of simulated annealing combined with redundant dihedral angle constraints were used to produce sets of converged structures with no significant restraint violations (distance and van der Waals violations < 0.2 Å and dihedral angle constraint violation < 5°), which were further refined in explicit solvent using the YASARA software with the YASARA forcefield [51]. The 30 HRP-2 IBD structures with the lowest total energy were selected, analyzed and validated using the Protein Structure Validation Software suite (http://psvs-1_5-dev.nesg.org). The constraints and structural quality statistics for the final set of water-refined HRP-2 IBD structures is summarized in Appendix A. The structure, NMR constraints, and resonance assignments were deposited in the Protein Data Bank (PDB, accession number 6T3I) and Biological Magnetic Resonance Bank (BMRB, accession number 34442).

In titration experiments 20 µM of ^15^N-labeled HRP-2_469–549_ or LEDGF/p75_345–426_ were mixed with various concentrations of unlabeled synthetic MLL peptide (123–160) or DMSO as a control. For each titration point, the chemical shift perturbations (CSP) in ^15^N/^1^H HSQC spectra measured in the SOFAST fashion were calculated and the dissociation constant was determined by a non-linear least squares analysis using GraphPad Prism and the equation
$${\mathrm{CSP}}_{obs}={\mathrm{CSP}}_{max}\times \frac{\left[L\right]+\left[P\right]+K_{D}-\sqrt{{\left(\left[L\right]+\left[P\right]+K_{D}\right)}^{2}-4\;\times \left[L\right]\times \left[P\right]}}{2\;\times \;\left[P\right]}$$
where CSP*_obs_* is the observed CSP at the given total ligand concentration [L], CSP*_max_* is the CSP at saturation, and [P] is the total concentration of protein [52]. 

### 2.7. HRP-2 Knockout Mouse Mode

C57BL/6N-Hdgfrp2<tm1b (KOMP)Wtsi>/Tcp (*HRP-2*^tm1b^) mice were ordered at Toronto Centre for Penogenomics after they were generated as part of the NorCOMM2 phenotyping project [53]. All animal experiments were approved by the KU Leuven ethical committee (P201/2014).

### 2.8. Clonogenic Growth In Vitro

Primary murine lin^-^ cells were cultured and scored in a semisolid colony formation unit (CFU) assay as described before [12]. Human, immortalized cell lines were cultured in MethoCult^TM^ H4230 (STEMCELL Technologies) and scored after 10 days in culture.

### 2.9. RNA-Sequencing and Bioinformatics

Total RNA samples (500 ng) were cleaned using the DNAse I kit (Thermoscientific) according to the Rapid out removal DNA kit instruction and converted into cDNA by using the QuantSeq 3ʹ mRNA-seq reverse 4 Library Prep Kit (Lexogen) according to manufacturer’s instructions [54] to generate a compatible library for Illumina sequencing. Briefly, library generation was initiated by oligodT priming for first strand cDNA which generated one fragment per transcript. The second strand cDNA was subsequently synthesized using random primers. Illumina-specific linker sequences were introduced by the primer with barcoding indices for different samples. The quality of cDNA libraries was determined using a High Sensitivity DNA Assay 2100 Bioanalyzer (Agilent) for quality control analysis. Sequencing of the cDNA library with 75bp single end reads was performed using an Illumina NextSeq 500 system. Reads were aligned to the reference genome GRCm38 using STAR-2.4.2a with default settings [55]. STAR was also used for gene expression quantification on the Ensembl GTF file version 84. Differential expression analysis was performed using DESeq2 in R [56]. The RNA-sequencing data are available in NCBI’s Gene Expression Omnibus (GSE154202).

### 2.10. Statistical Analysis

The presented graphs were generated using GraphPad Prism 8.3.1. The experimental results from in vitro experiments are presented as means ± standard deviations. Sample sizes (*n*) and description of each experimental group or condition are indicated in the figure legends. Different groups were statistically evaluated by a two-sided student’s t-test (for *n* > 4) or Mann–Whitney U test (for *n* = 4) using GraphPad Prism 8.3.1 and *p*-values below 0.05 were considered significantly different.

## 3. Results

### 3.1. HRP-2 Interacts with MLL in the Absence of Menin

HRP-2 and LEDGF/p75 are the only two human proteins that contain both the PWWP and the IBD domains (Figure 1A) [15,33]. As both HRP-2 and LEDGF/p75 can fulfil a similar role in HIV infection [33,37] and LEDGF/p75 is important for *MLL-*r [9,12], we investigated the involvement of HRP-2 in hematopoiesis and MLL-mediated transformation. Menin serves as an adaptor to link MLL with LEDGF/p75 and thus is essential for the formation of the triple transcription-regulatory complex [9]. Here, we investigated the potential interaction of HRP-2 with the MLL-menin complex by in vitro co-immunoprecipitation (IP) in HEK293T cells transfected with flag-tagged MLL-ELL and/or menin-HA expression constructs (Figure 1B–E). The binding of endogenous HRP-2 and LEDGF/p75 was detected using anti-HRP-2 or anti-LEDGF antibodies. Despite the poor detection of flag-MLL-ELL in the precipitate in the absence of ectopic menin expression, it is clear that upon the overexpression of menin and MLL, both HRP-2 and LEDGF/p75 were precipitated. To further clarify whether the binding of HRP-2 to MLL-ELL is dependent on menin, as is the case for LEDGF/p75, we treated cellular lysates with 100 µM of MI-2 [57], a previously described MLL-menin interaction inhibitor. In line with previous reports, MI-2 treatment resulted in a partial loss of menin binding upon MLL precipitation (Figure 1C) [57]. HRP-2 still co-precipitated with MLL upon MI-2 treatment. Similar results were obtained using 50 µM of the more potent MI-538 inhibitor (Figure 1D) [58]. Since both menin inhibitors did not fully abrogate the MLL-menin interaction, we introduced three point mutations (F9A, P10A and P13A) into a flag-MLL-ELL_1–330_ (MLL Mut), known to completely abolish binding to menin [59]. Whereas the MLL mutations did not interfere with the precipitation of HRP-2 (Figure 1E), the MLL mutant failed to co-IP LEDGF/p75 despite the presence of overexpressed menin, suggesting HRP-2 is less dependent on menin for binding to MLL. Interestingly, whereas ectopic overexpression of menin resulted in increased co-precipitation of LEDGF/p75, binding to HRP-2 was reduced, supporting a competition between HRP-2 and LEDGF/p75 for binding to MLL under control of menin (Figure 1B,E). 

### 3.2. HRP-2 and LEDGF/p75 Interact with MLL through a Conserved Interface with Similar Affinities 

To obtain a detailed insight into the mechanism of the MLL and HRP-2 association, we determined the solution structure of the HRP-2-IBD domain (amino acids 469–549, Figure 2A). The solution structure of the HRP-2-IBD revealed a compact right-handed bundle composed of five α-helices, comparable to other members of the TFIIS N-terminal domain family and demonstrated a high degree of structural conservation between LEDGF/p75 and HRP-2 IBDs (Figure 2C). 

To validate and characterize the direct interaction between HRP-2-IBD and MLL, we followed the changes in positions of backbone NMR signals of ^15^N-labeled HRP-2-IBD either in absence or presence of different concentrations of a synthetic MLL-derived peptide (amino acids 123-160). MLL_123–160_ induced significant chemical shift perturbations of the IBD backbone signals (Figure 2B and Appendix A). Moreover, this experiment revealed that MLL recognizes HRP-2-IBD through the same interface and binds with similar affinity as LEDGF/p75 (Figure 2D–H). In particular, the chemical shift perturbations in the IBD backbone induced by binding of MLL_123–160_ were found in two regions (amino acids 479–492 and 520–535) (Figure 2D). As for the LEDGF/p75 IBD-MLL_123–160_ interaction [46], these regions form two interhelical loops connecting IBD α helices α1–α2 and α4–α5, respectively (Figure 2E,F). Additionally, the analysis of the chemical shift perturbations in the HRP-2 IBD backbone induced by binding of JPO2 (amino acids 1–130) and POGZ (amino acids 1117–1410) revealed a pattern remarkably similar to that induced upon addition of MLL confirming that these protein fragments bind to HRP-2 in the same conserved structural mode as LEDGF/p75 IBD (Appendix A). Importantly, the dissociation constants for the MLL_123–160_ interaction with HRP-2-IBD (54.4 ± 2.2 µM, Figure 2G) obtained from NMR titration experiments are comparable with those of LEDGF/p75 (64.0 ± 6.5 µM) obtained in our earlier studies [30]. In addition, HRP-2 and LEDGF/p75 affinities to MLL were compared in an AlphaScreen assay with MBP-fused recombinant C-terminal fragments of LEDGF/p75 (LEDGF_325–530_) and HRP-2 (HRP-2_470–552_), purified from E. *coli* and used to outcompete the interaction between recombinant flag-tagged LEDGF/p75 and GST-tagged N-terminal fragment of MLL (MLL_1–160_–GST). Unlike MBP alone, both IBD domains of HRP-2 and LEDGF/p75 efficiently outcompeted the interaction between MLL_1–160_–GST and full-length flag-LEDGF/p75 (Figure 2H). Altogether, our data revealed that the overall binding mechanism used by HRP-2 and LEDGF/p75 IBDs is highly conserved.

### 3.3. Systemic HRP-2 Depletion in Mice Leads to Increased Postnatal Mortality and Decreased In Vitro Colony Formation of Hematopoietic Stem Cells

To better understand the importance of the MLL-HRP-2 interaction, we investigated a systemic knockout mouse model (from the Toronto Centre Phenogenomic, Toronto, ON, Canada) to address the role of HRP-2 in postnatal hematopoiesis. The depletion of HRP-2 mRNA and protein levels in these knockout (*HRP-2*^−/−^) mice were confirmed by quantitative PCR (qPCR) and Western blot in bone marrow cells (Appendix A). With percentages of 61% and 29.5%, observation of inbred crossings revealed increased numbers of heterozygous (*HRP-2^+/−^*) and wild type (*HRP-2*^+/+^) mice, respectively, at age of 6 to 8 weeks in regard to the expected Mendelian inheritance pattern (Table 1). Moreover, *HRP-2^−/−^* mice were present at a lower percentage (9.5%) than expected. At one day after birth, the ratio of *HRP-2^−/−^* mice corresponded more closely to the expected Mendelian ratio by representing 20% of newborn pups.

In contrast to an earlier report by Wang et al. [61], we observed that *HRP-2^−/−^* pups presented with increased mortality before 6–8 weeks of age, suggesting that HRP-2 is important for postnatal survival early after birth. Read-through of the gene trap could explain discrepancies between both models, since Wang et al. showed a 5–20% residual *HRP-2* expression in their model, while *HRP-2* mRNA levels in hematopoietic stem cells (HSC) of our few surviving mice were undetectable by qPCR (Appendix A).

To compare steady-state hematopoiesis between weaned wild type and knockout mice, blood counts were analyzed for the different genotypes. No significant differences were observed in total white and red blood cell count (Figure 3A top), whereas the differential blood count revealed a significant increase in neutrophils in the *HRP-2*^−/−^ mice compared to the wild type (*p* = 0.042, Figure 3A bottom). Other cell types did not differ between genotypes (Appendix A). To explore the functionality of the cells, we sought to compare the colony-forming capacity of *HRP-2* wild type, heterozygous and knockout cells using myeloid CFU assays. Lineage depleted (lin^-^) cells were harvested from *HRP-2^+/+^*, *HRP-2^+/−^* and *HRP-2^−/−^* mice and serially plated. After two rounds of plating, the number of colonies derived from *HRP-2^−/−^ and HRP-2^+/−^* bone marrow cells were respectively 80% and 33% lower compared to the wild-type control (Figure 3B).

To gain more detailed knowledge of the *HRP-2* knockouts, we performed a gene expression profile analysis, comparing RNA of lin^−^ bone marrow cells of *HRP-2*^+/+^ and *HRP-2*^−/−^ mice. We found a total of 52 differentially expressed genes (FDR 0.25, Appendix A), of which 23 were upregulated and 29 were downregulated in *HRP-2*^−/−^ cells. Gene set enrichment analysis (GSEA) uncovered that *HRP-2*^−/−^ cells display a trend towards a gene signature of myeloid differentiation (*p* < 2.2 × 10^−16^, Figure 3C). Moreover, the downregulation of interferon alpha and gamma pathways was observed in multiple gene sets (*p* < 2.2 × 10^−16^, Figure 3C).

Taken together, these results suggest that HRP-2 depletion reduces the colony formation capacity and induces myeloid differentiation, suggesting that HRP-2 is involved in maintaining the stem-like state of bone marrow cells.

### 3.4. HRP-2 Depletion Impairs the Clonogenic Growth of Both Human and Mouse Leukemic Cell Lines Independently of MLL Fusions

Next, we investigated the role of HRP-2 in leukemic transformation induced by oncogenic MLL fusions in human leukemic cell lines harboring the MLL-fusions MLL-AF9 (THP1) or MLL-AF4 (SEM), as well as the MLL wild-type leukemic cell lines Kasumi1, K562, and Nalm6. All cell lines were transduced with a lentiviral vector to deplete HRP-2 or with a control vector and plated in methylcellulose. Reduced HRP-2 mRNA and protein levels were verified by qPCR and western blot (Appendix A). Of note, LEDGF/p75 mRNA and protein levels remained unaffected upon HRP-2 depletion (Appendix A). After 12 days, a decrease in the number of colonies of THP1 (32.8% ± 2.1%) and SEM (51.5% ± 7.0%) cells was observed (Figure 4A, left). Interestingly, the number of colonies in MLL germline cell lines also decreased upon HRP-2 knockdown (Figure 4A, right) with 44.4% ± 27.4% (K562), 38.3% ± 14.2% (Kasumi1) and 19.3% ± 8.8% (Nalm6). Experiments with higher vector titers resulted in an even more pronounced drop in the number of colonies (Figure 4 inserted table), suggesting a concentration-dependent effect. In liquid culture, an impaired cellular growth was observed for SEM and Kasumi1 cells but not for THP1, K562 and Nalm6 cells, excluding that impaired cell growth by HRP-2 depletion is affected in an *MLL*-r-dependent way (Appendix A). Also, for cultured murine lin^-^ bone marrow cells expressing an MLL-AF9 fusion or a control fusion E2A-HLF, respectively 70% and 56% less colonies were observed after a lentiviral-induced *HRP-2* knockdown (Figure 4B). Taken together, these observations suggest that loss of HRP-2 generally impairs growth of human and murine leukemic cells even in the absence of MLL fusions. To further emphasize that the phenotype observed was independent of MLL, HoxA9 levels were quantified by qPCR (Appendix A). HoxA9 levels were not significantly affected by HRP-2 depletion.

### 3.5. HRP-2 Overexpression Rescues MLL-r Clonogenic Growth in LEDGF/p75-Depleted Cells

Since HRP-2 depletion affects the clonogenic growth of all leukemic cell lines tested, we investigated whether HRP-2 overexpression could rescue the colony-forming capacity of an MLL-AF9 leukemic cell line (THP1) after LEDGF/p75 depletion. First, we stably expressed miRNA-resistant LEDGF/p75, HRP-2 or a mock control. Subsequently, cell lines were transduced with a miRNA-based lentiviral vector to specifically knockdown *Psip1* or eGFP (Figure 4C). Expression levels of *Psip1* and *HRP-2* in the generated cell lines were verified by qPCR and western blot (Appendix A). In line with previous reports, LEDGF/p75 depletion caused a ~65% decrease in the number of colonies formed compared to the control [9,12,45] (Figure 4C) and this defect was rescued by LEDGF/p75 back-complementation (BC). Of interest, overexpression of HRP-2 fully restored the CFU activity to wild-type levels, indicating that overexpression of HRP-2 can functionally compensate for the absence of LEDGF/p75 in MLL-transformed cells. This experiment was repeated in the MLL wild-type Nalm6 cell line (Appendix A). In line with published data [12,45], the depletion of LEDGF/p75 did not affect the colony forming capacity of the Nalm6 cell line, as the LEDGF/p75-dependent drop in colonies is MLL-specific. Although both HRP-2 and LEDGF/p75 overexpression increased the colony formation of Nalm6 cells, the effect was not significant.

### 3.6. HRP-2 Is Dispensable for the Initiation of MLL-r Leukemia

Our data imply that HRP-2 and LEDGF/p75 own physiological different roles in cells. Therefore, we investigated the involvement of HRP-2 in the initiation of *MLL*-r leukemia. HSC harvested from *HRP-2^+/+^*, *HRP-2^+/−^* and *HRP-2^−/−^* mice were transduced with lentiviral vectors encoding one of the most common MLL-fusion proteins (MLL-ENL) or a control fusion inducing ALL (E2A-HLF) and their transformation potentials were compared in serial plating assays (Figure 5A). Remarkably, *HRP-2* wild type, heterozygous, and knockout cells were efficiently transformed by MLL-ENL and E2A-HLF as revealed by the increased number of colonies after three rounds in the CFU assay (Figure 5B). The drop of colonies in the second round in MLL-ENL is likely due to a low transduction efficiency and represents a selection step prior to transformation. Finally, we performed an in vivo BMT assay. MLL-ENL–expressing *HRP-2^+/+^*, *HRP-2^+/−^* and *HRP-2^−/−^* cells were transplanted into lethally irradiated recipient mice and monitored for leukemogenesis. Kaplan-Meier survival plots reveal that mice transplanted with MLL-transformed *HRP-2^+/−^* and *HRP-2^−/−^* cells died significantly faster than the transformed wild-type cells (*p* = 0.02 and *p* = 0.006, respectively) (Figure 5C), excluding the requirement of HRP-2 in leukemogenesis.

## 4. Discussion

Mixed lineage leukemia-rearranged (*MLL*-r) leukemia, a genetically distinct subset of leukemia characterized by *MLL*-rearrangements, is an aggressive form of leukemia without specific treatment options. Currently, many efforts are ongoing to specifically target the oncogenic multi-protein complex involved in *MLL*-r. DOT1L and BET inhibitors target the MLL fusion partners [62,63], whereas menin inhibitors are directly inhibiting MLL binding [64].

The PWWP domain of LEDGF/p75 recognizes H3K36me3 marks associated with active transcription, while the IBD domain binds cellular proteins such as MLL and oncogenic MLL-fusions [46]. MLL fusions are tethered to the chromatin via LEDGF/p75 (encoded by the *Psip1* gene) and interference with this tethering mechanism provides an additional potential target to treat *MLL*-r [12]. Since HRP-2, a paralog of LEDGF/p75 with high sequence similarity, also contains both structured domains and functionally compensates for LEDGF/p75 in the context of HIV [37], we investigated whether HRP-2 is involved in MLL fusion-induced leukemia.

To date, little is known about the function of HRP-2 in cell biology and oncogenesis. Co-immunoprecipitation experiments indicated PogZ, JPO2, and IWS as HRP-2 binding partners [27,31]. We confirmed binding of PogZ and JPO2 to HRP-2 by NMR (Appendix A). In addition, we show for the first time that MLL interacts with the IBD of HRP-2 (Figure 1 and Figure 2 and Appendix A). The role of menin in the MLL-complex has been described before [10] and menin inhibitors induce an MLL specific effect [65]. Whereas menin was required for the stabilization of the MLL-LEDGF/p75 interaction as shown before [9], co-IP experiments with selective MLL-menin interaction inhibitors (Figure 1C,D) and menin-deficient MLL mutants (Figure 1E) revealed that HRP-2 is less dependent on menin. Despite similar binding affinities as measured by NMR and AlphaScreen for the direct binding of HRP-2 or LEDGF/p75 to MLL (Figure 2G,H), co-IP experiments indicate that menin modulates the interaction between the IBD and MLL in favor of LEDGF/p75 (Figure 1). We hypothesize that either low (undetectable) menin levels are sufficient to support this interaction or addition of menin increases the binding affinity of LEDGF/p75 for MLL at the expense of HRP-2. Alternatively, it is possible that the HRP-2-MLL interaction is differentially regulated by PTMs or another cellular factor as compared to the LEDGF/p75-HRP-2 interaction.

We investigated the role of HRP-2 in hematopoiesis in a HRP-2 knockout mouse model. In contrast to Wang et al. [61], the offspring from a heterozygous HRP-2 (HRP-2^+/^) breeding couple deviated from the expected Mendelian inheritance pattern (Table 1), suggesting that HRP-2 is important for postnatal survival. Of note, high postnatal lethality was also observed in systemic Psip1 knockout mice with less than 1% reaching weaning age [66]. Unlike the phenotype observed in Psip1 knockout mice [66], we can only distinguish adult HRP-2 knockout mice from their littermates by small differences in the hematopoietic system compared to wild type and heterozygous littermates (Figure 3). Colony formation experiments (Figure 3B) and Gene Set Enrichment Analysis (Figure 3C) on lin^-^ HSC hinted towards a stem-like state supported by HRP-2. Recent studies indicated a crucial function of HRP-2 during mRNA transcription [39] and differentiation [67] of muscle cells, suggesting HRP-2^−/−^ muscles fail to support vital functions.

In addition, an analysis of cell survival (Figure 4 and Appendix A) indicates that HRP-2 depletion impairs cellular growth of different types of leukemia, independently of the presence of MLL fusions. Of interest, *HRP-2* knockdown was reported to reduce growth in hepatocellular carcinoma cells and induce cell death in U2OS cells [38], suggesting a more general pro-survival role of HRP-2. Of note, we observed that colony formation of the MLL-driven cell lines THP1 and SEM at high vector titer were less affected by HRP-2 depletion than the MLL wild-type cell lines K562, Kasumi1 and Nalm6 (Figure 4). We hypothesize that HRP-2 and LEDGF/p75 compete for MLL binding, implying that reduced HRP-2 levels facilitate binding between LEDGF/p75, MLL and menin supporting *MLL-*r-induced leukemogenesis.

In fact, while endogenous HRP-2 levels do not rescue colony formation in THP1 cells upon depletion of LEDGF/p75, overexpression of HRP-2 does (Figure 4C). This latter result supports the notion that HRP-2 can function as a tether between the MLL-fusion proteins and its target genes. However, as HRP-2 overexpression also promoted cell growth in hepatocellular carcinoma [31], we cannot exclude that HRP-2 rescues the cellular growth via a more general pathway. This rescue phenotype is reminiscent of that of HRP-2 in HIV infection (25).

Although HRP-2 seems to play an important but nonspecific role in leukemic cell survival, our results indicate that HRP-2 is not important for the initial transformation of hematopoietic stem cells by MLL fusions. MLL-ENL transduced lin^-^ BMC from *HRP-2* knockout, heterozygous or wild-type mice transformed irrespective of the genotype in a colony forming assay, indicating that *HRP-2* knockdown, nor knockout is impairing MLL-driven leukemogenesis in the presence of LEDGF/p75.

In an in vivo experiment in lethally irradiated mice, engrafted MLL-ENL transduced cells induced MLL leukemia with a survival time of 6 weeks post transplantation (Figure 5C), equal to previously published experiments [12]. Transplantation of MLL-ENL transduced HRP-2^−/−^ or HRP-2^+/−^ cells also resulted in leukemogenesis and the life span of these mice was even shorter. The shorter life span of mice in the absence of HRP-2 may be due to a loss of competition between HRP-2 and LEDGF/p75 for binding to MLL. By competing with LEDGF/p75, HRP-2 could act as a tumor suppressor. Alternatively, although HRP-2 might be dispensable for leukemogenesis, its role in hematopoiesis may affect the survival of mice.

## 5. Conclusions

In conclusion, our results demonstrate that although HRP-2 interacts with MLL, this interaction is not required for the development of mixed lineage leukemia. LEDGF/p75 is the main driver of *MLL-*r-driven leukemia and menin, a modulator of the relative affinities for MLL, may be responsible for this selectivity by increasing the affinity of MLL for LEDGF/p75. Still, HRP-2 may have a more general role in the survival of leukemic cells independently of MLL.

## Figures and Tables

**Figure 1 cells-10-00192-f001:**
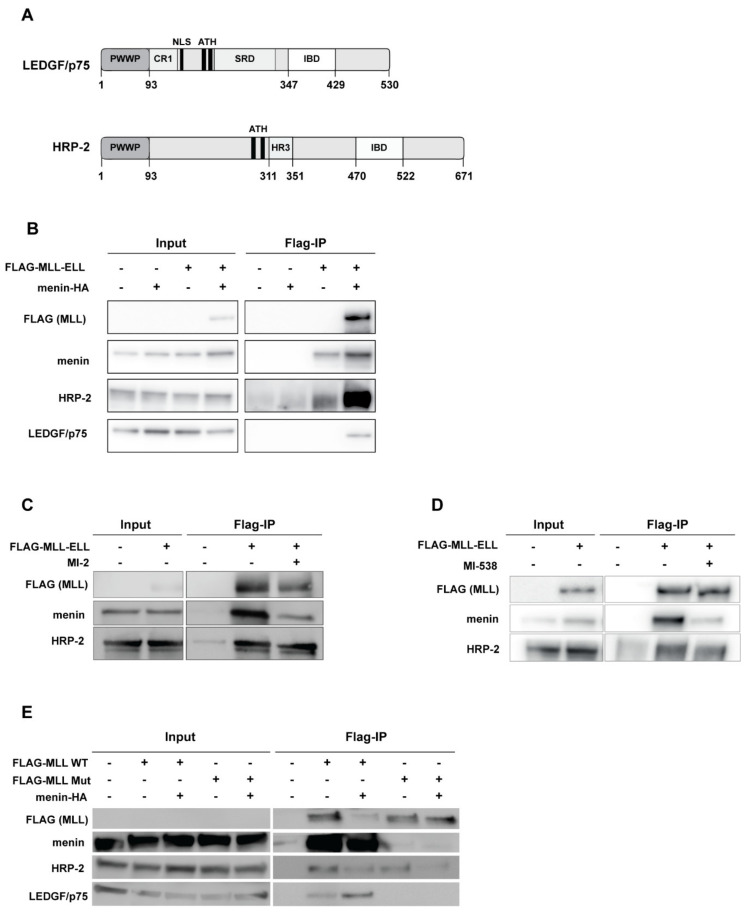
Menin is dispensable for the interaction of MLL and HRP-2. (**A**) Schematic representation of the domain structure of LEDGF/p75 and HRP-2. The N-terminus of LEDGF comprises the PWWP domain, the nuclear localization signal (NLS), AT hooks (ATH) and four charged regions (CR), of which CR2-4 contain a supercoiled DNA recognition domain (SRD). At the C-terminus, LEDGF/p75 harbors the integrase binding domain (IBD) [60]. HRP-2 harbors an N-terminal PWWP domain, two AT hooks, a patch of conserved amino acids known as homology region III (HR3) and a C-terminal IBD domain; (**B**) HEK293T cells were transfected with flag-tagged MLL-ELL and/or menin-HA expression constructs as indicated. Flag-MLL-ELL was immunoprecipitated using anti-flag beads and analyzed using antibodies against flag. Menin, endogenous HRP-2 and LEDGF/p75 were detected using specific antibodies; (**C**,**D**) HEK293T cells were transfected with flag-tagged MLL-ELL as indicated. MLL-ELL was immunoprecipitated using anti-flag beads in the presence of a previously described menin-MLL interaction inhibitor (**C**) MI-2 [57] or (**D**) more potent MI-538 [58], at concentrations of 100 and 50 µM respectively or DMSO control. Precipitated proteins were analyzed by western blot. Flag antibodies were used for the detection of MLL-ELL. Endogenous levels of menin, HRP-2 and LEDGF/p75 were detected using specific antibodies; (**E**) HEK293T cells were transfected with flag-tagged wild-type MLL-ELL_1–330_ (MLL WT) or a menin interaction-deficient construct ‘MLL Mut’ with point mutations F9A, P10A and P13A and/or menin-HA as indicated. Due to low expression levels, MLL_1-330_ is not detected in the input by flag antibody. Endogenous levels of menin, LEDGF/p75 and HRP-2 are detected using specific antibodies. Details about the western blot analysis can be found in supplementary information.

**Figure 2 cells-10-00192-f002:**
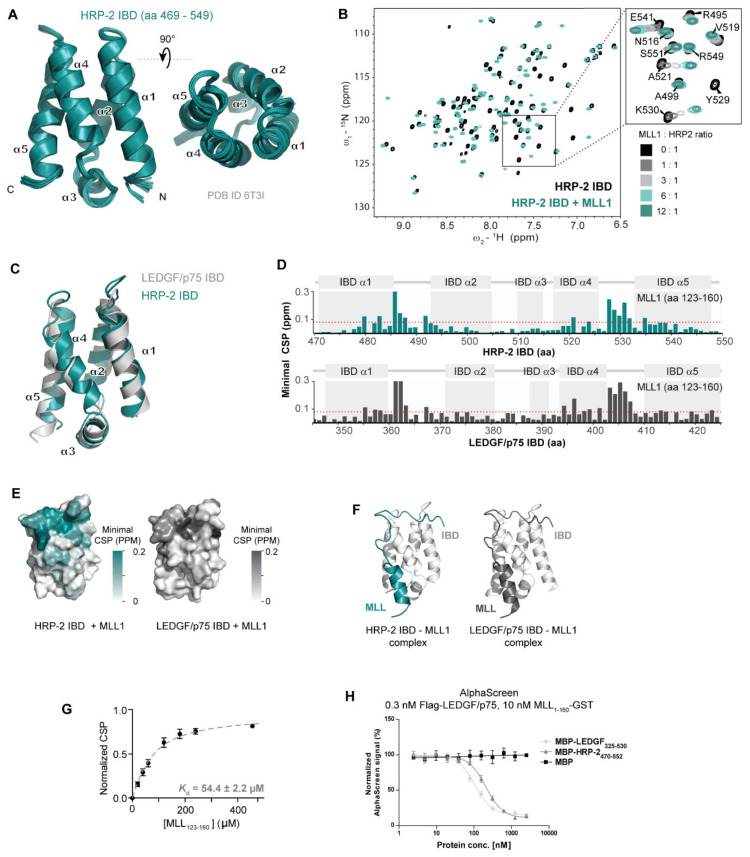
HRP-2 IBD and LEDGF/p75 IBD interact with MLL in a conserved manner. (**A**) Solution structure of HRP-2 IBD (PDB ID 6T3I); (**B**) HRP-2-IBD directly interacts with the identical consensus motif of MLL (amino acids 123-160) alike LEDGF/p75-IBD as determined by NMR spectroscopy. Comparison of the 15^N^/1^H^ HSQC spectra of the 20 µM HRP-2 IBD in the absence (black) and presence (green) of 120 µM MLL_123–160_. On the right, detail HRP-2 IBD titration with MLL_123–160_. HSQC spectra are colored based on MLL_123–160_ concentration as indicated in the figure. The spectra were obtained from the 15^N^-labeled recombinant IBD and the unlabeled synthetic MLL-derived peptide; (**C**) Superposition of HRP-2-IBD (green) and LEDGF/p75-IBD (light grey) solution structures; (**D**,**E**) Comparison of the HRP-2-IBD–MLL_123–160_ and LEDGF/p75-IBD –MLL_123–160_ interaction surfaces. Representation of the minimal chemical shift perturbation (CSP) in backbone amide signals of the IBDs upon addition of MLL_123–160_ peptide in panel D. Amino acid residues that are significantly perturbed upon addition of MLL_123–160_ to HRP-2-IBD or LEDGF/p75-IBD (as determined by NMR spectroscopy) are highlighted in green or gray on the surface of the IBD structures in panel E; (**F**) Comparison of homology model of MLL-HRP-2-IBD and solution structure of MLL-LEDGF/p75-IBD solution structure (PDB ID 6emq); (**G**) *K*_d_ fit from NMR titrations of HRP-2-IBD with MLL_123–160_. Dissociation constant was determined by following the chemical shift perturbations of the HRP-2-IBD backbone amide signals induced upon titration with MLL_123–160_. Error bars represent the error of the fit for most perturbed residues (*n* = 10); (**H**) Alpha Screen. Full length flag-tagged LEDGF/p75 (0.3 nM) was incubated with GST-tagged MLL_1–160_ (10 nM) in an out competition Alpha Screen assay. C-terminal fragments of LEDGF/p75 (LEDGF_325–530_) and HRP-2 (HRP-2_470–552_) were titrated to outcompete the interaction.

**Figure 3 cells-10-00192-f003:**
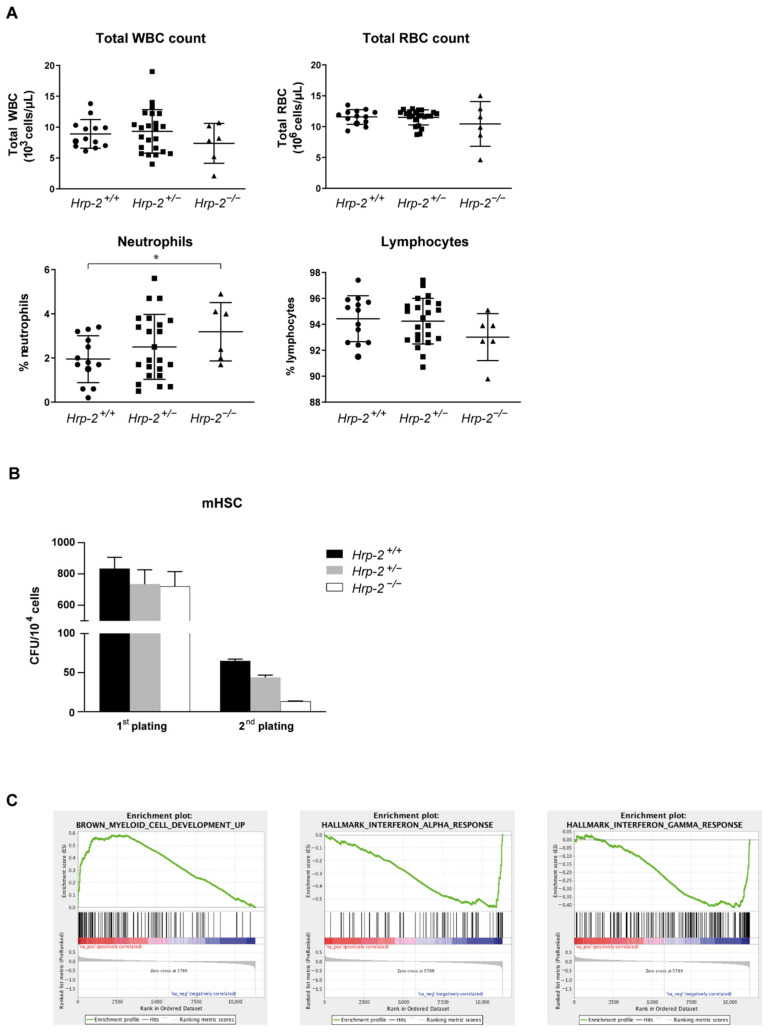
*HRP-2* knockout mice show subtle hematopoietic defects. (**A**) Peripheral blood counts in wild type (*HRP-2*^+/+^), heterozygous (*HRP-2*^+/−^) and knockout (*HRP-2*^−/−^) mice analyzed for total with blood cell (WBC) and total red blood cell (RBC) count (top). Level of neutrophils and lymphocytes (below) are presented as percentage of total differential white blood cell count. Average and standard deviation are indicated. Significance level was determined using two-sided student’s *t*-test (* *p* = 0.042); (**B**) Number of colonies for 10^4^ lineage depleted cells harvested from *HRP-2*^+/+^, *HRP-2*^+/−^, and *HRP-2*^−/−^ mice in two consecutive platings in a myeloid CFU assay. Error bars indicate standard deviations of duplicate measurements; (**C**) Gene set enrichment analysis (GSEA) showing a correlation between the RNA profile of *HRP-2^−/−^* lineage depleted (lin^-^) bone marrow cells compared to the gene signature of myeloid differentiation and down-regulation of interferon pathways alpha (middle) and gamma (right).

**Figure 4 cells-10-00192-f004:**
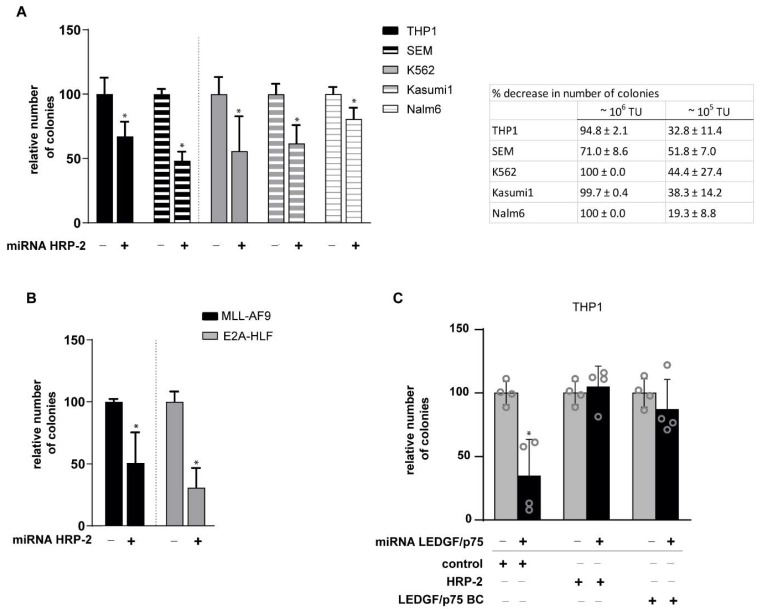
*HRP-2* knockdown impairs growth in leukemic cell lines, whereas *HRP-2* overexpression rescues the leukemic phenotype. (**A**) Human MLL-transformed cell lines THP1 (MLL-AF9) and SEM (MLL-AF4), as well as wild type MLL cells (Nalm6, Kasumi1, and K562), separated by the dotted line, were transduced with a lentiviral vector encoding a miRNA to knockdown *HRP-2* or a control vector (-). After 12 days in methylcellulose the number of colonies was scored. Counts were normalized to their associated control. Error bars indicate standard deviations of four replicates. Differences were determined using Mann-Whitney U test; * *p* < 0.05. Inserted table describes average percentage decrease in number of colonies for indicated vector titers. TU = titer units (p24 pg/mL) ± S.D.; (**B**) Colony forming assay (CFU) after *HRP-2* knockdown or control (-) of primary bone marrow cells harvested from leukemic mice transplanted with MLL-AF9 or E2A-HLF transduced cells. Counts were normalized to their associated control. Error bars indicate standard deviations of four replicates. Differences were determined using Mann-Whitney U test; * *p* < 0.05; (**C**) Relative number of colonies per 500 plated cells for the THP1 cell line overexpressing miRNA resistant LEDGF/p75, HRP-2 or empty vector (control) after transduction with a lentiviral vector expressing a LEDGF/p75-miRNA to knockdown LEDGF/p75 or a control (black -). Error bars indicate standard deviations of four measurements, individually indicated by a circle. Differences were determined using Mann-Whitney U test; * *p* < 0.05.

**Figure 5 cells-10-00192-f005:**
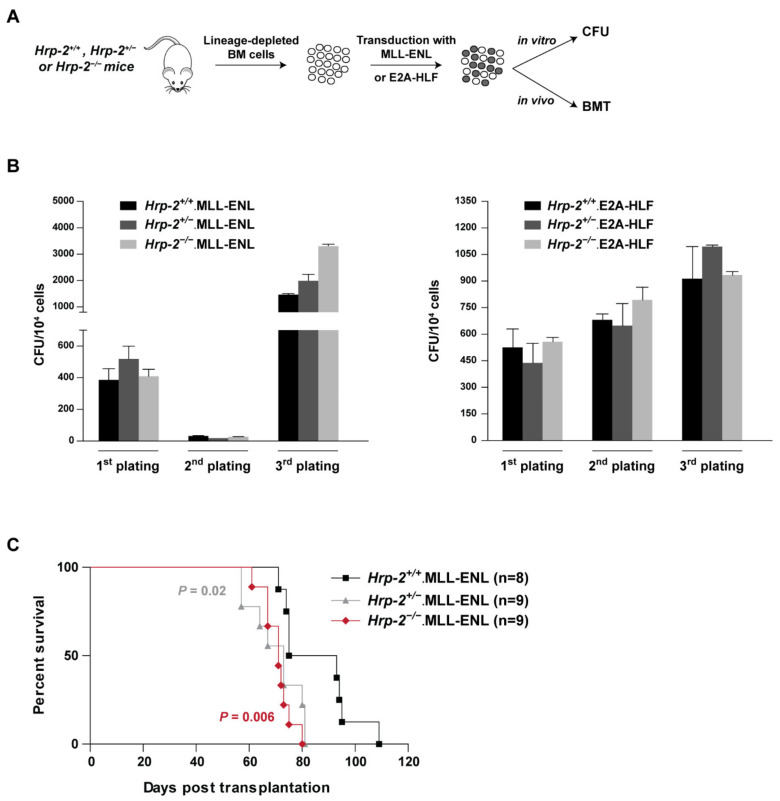
HRP-2 is not required for the initiation of leukemia in vivo. (**A**) Schematic representation of the experimental set up. BM = bone marrow; CFU = colony forming assay; BMT = bone marrow transplantation; (**B**) Colony-forming assay (CFU) for 10^4^
*HRP-2^+/+^*, *HRP-2^+/−^* and *HRP-2^−/−^* cells transduced with a retroviral vector encoding the MLL-ENL (**B**, left) or E2A-HLF (B, right) fusion. Error bars represent standard deviation of triplicate measurements; (**C**) Kaplan-Meier survival curve from bone marrow transplantation experiments of irradiated mice who received *HRP-2* wild type (^+/+^), heterozygous (^+/−^) or knockout (^−/−^) cells, transduced with mouse stem cell virus (MSCV) MLL-ENL expression vector. Number of transplanted animals (*n*) per group in indicated. Statistical differences were determined using GraphPad prism.

**Table 1 cells-10-00192-t001:** Genotype of heterozygous (*HRP-2^+/−^*) crossed offspring after birth or weaning (6 to 8 weeks).

	Number (%) of Mice		
Age	*HRP-2^+/+^*	*HRP-2^+/−^*	*HRP-2^−/−^*
1 day	36 (23.2)	88 (56.7)	31 (20.0)
6–8 weeks	31 (29.5)	64 (61.0)	10 (9.5)
Expected values	25%	50%	25%

## Data Availability

The RNA sequencing dataset generated and analyzed during the current study is available in the NCBI’s Gene Expression Omnibus repository (GSE154202). Generated NMR structures are available on the PDB (ID: 6T3I) and BMRB (ID: 34442). Additional data generated in this study is available from the corresponding author Zeger Debyser (Zeger.Debyser@kuleuven.be).

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
