# Peer review of "Unlike its Paralog LEDGF/p75, HRP-2 Is Dispensable for MLL-R Leukemogenesis but Important for Leukemic Cell Survival"

_cells, 2021, doi:10.3390/cells10010192_

Round 1

Reviewer 1 Report

The manuscript details the molecular background of the interaction of HRP-2 and MLL1, with respect to the highly similar LEDGF/p75 protein. The question is relevant, since in the family of HDGF proteins only HRP2 shares the MLL-menin-interacting IBD with LEDGF/p75. The hypothesis that HRP-2 binds MLL1 in a similar fashion as LEDGF/p75 is well-founded and worth investigating. The authors applied various techniques from binding studies, structure determination, cellular and in vivo experiments to study the molecular background of HRP-2-MLL interaction and its physiological relevance. With the combination of this wide variety of different methods, they were able to point out important differences between HRP-2 and LEDGF/p75. One of their main finding is that HRP-2 is able to bind MLL1 directly in a very similar structure as LEDGF/p75, but without the need for menin, which is indispensable for the LEDGF-MLL interaction. The results presented in the manuscript clarify the role of HRP-2 in the normal haematopoiesis and suggest that HRP-2 helps maintaining the stem-like state of bone-marrow cells. They also show that HRP-2 can compensate for the loss of LEDGF/p75 in leukemic cell lines, but it is not necessary for the initiation of MLL-rearranged leukemia. These findings reveal important details of the function of HRP-2 and the regulation of haematopoiesis and leukemogenesis. The methods used are adequate and suitable and the findings are explained in sufficient detail to corroborate the conclusions of the authors. The figures are informative and clear, except for the ones where the small legends make reading more difficult.

I have only some minor suggestions:

  1. In the equation on page 6., CSPmax is used, while in the text it's termed CSPsat.
  2. The legends on Figures 1. and 3. are very small and difficult to read, I suggest to enlarge the fonts.

The results presented are convincing and the conclusions drawn are well-supported by the data. The discussion is thorough and details the different implicatons of the findings.

All in all, the manuscript describes a thorough and convincing work with important results of interest for a wide range of readers.

Reviewer 2 Report

Due to a similarity in protein domains between two proteins, LEDGF/p75 and HRP- 2, the authors question the implication of HRP- 2 in hematopoiesis and leukemogenesis.

They show that MLL interacts with the IBD domain of HRP-2 and that this interaction is not dependent on the presence of menin, on the contrary to LEDGF/p75. Thanks to a mouse model, the authors demonstrate the implication of HRP-2 in hematopoiesis. Finally, the authors address the role of HRP-2 in leukemogenesis. They demonstrated that HRP-2 depletion impairs cellular growth of different types of leukemia, independently of the presence of MLL fusions.

The rationale of the study is clear and convincing. The aim of the study demonstrates a certain novelty and interest. The experiments are well carried out, and the interpretations are in line with the results.

Major comments

  1. Mat and Meth/ Statistical analysis

When the number of replicates is low, two-sided student’s t-test is not the most appropriate test. Instead, non parametric tests are recommended. Please modify this in the text and the figures.

  1. Results

Figure 1B. The WB is dirty with irregular protein bands, undermining the credibility of the data. Please, provide a better WB.

Figure 1B. Please provide the blot with endogenous menin.

Figure 1D is not convincing, it seems HRP-2 is still present even in the first lane of the Co-IP. Please, provide a better WB.

Minor comments

Introduction:

  1. « Mixed Lineage Leukemia (MLL)-rearranged leukemia is an aggressive, genetically distinct subset of acute leukemia causing 70% of pediatric leukemias”. It is unclear whether the 70% are the MLL leukemia or the other leukemia. Reformulate.
  2. « Importantly, LEDGF/p75 is dispensable for normal hematopoiesis ». Please, if known, indicate the other physiological processes related to a normal function of LEDGF/p75.
  3. Indicate in the introduction whether HRP-2 is found mutated/amplified/lost/fused in leukemia context ? in other cancer ?

Mat and Meth:

  1. Co-IP : it is useless to indicate the quantity of plasmid if the total cell number is not indicated.
  2. Clonogenic growth in vitro
 : indicate mouse for the lin - cells.
